# Long Context Compression with Activation Beacon

**Peitian Zhang**[1,2*] **Zheng Liu**[1†] **Shitao Xiao**[1]  **Ninglu Shao**[1,2]  **Qiwei Ye**[1]  **Zhicheng Dou**[2]
1: Beijing Academy of Artificial Intelligence,
2: Gaoling School of Artificial Intelligence, Renmin University of China
`{namespace.pt, zhengliu1026}@gmail.com`

## Abstract

Long context compression is a critical research problem due to its significance in reducing the high computational and memory costs associated with LLMs. In this paper, we propose Activation Beacon, a plug-in module for transformer-based LLMs that targets effective, efficient, and flexible compression of long contexts. To achieve this, our method introduces the following technical designs. 1) We directly compress the activations (i.e. keys and values at every layer), rather than leveraging soft prompts to relay information (which constitute a major bottleneck to encapsulate the complex information within long contexts). 2) We tailor the compression workflow, where each fine-grained input unit is progressively compressed, enabling high-quality compression and efficient computation during both training and inference. 3) We train the model through compression-based auto-regression, making full use of plain texts and instructional data to optimize the model's compression performance. 4) During training, we randomly sample a compression ratio at each step, teaching the model to support a wide range of compression configurations. Extensive evaluations are conducted on various long-context tasks whose lengths (e.g., 128K) may far exceed the maximum training length (20K), such as document understanding, few-shot learning, and Needle-in-a-Haystack. Whilst existing methods struggle to handle these challenging tasks, Activation Beacon maintains a comparable performance to the uncompressed baseline across various scenarios, achieving a 2x acceleration in inference time and an 8x reduction of memory costs for KV cache.

## 1 Introduction

Large language models (LLMs) need to process long contexts to accomplish many important tasks, such as long-document understanding (Jiang et al., 2024b), long-content creation (Bai et al., 2024), and long-term memorization/reasoning (Zhang et al., 2024). To address these needs, modern LLMs are built with extended context windows (e.g., 128K) that enable remarkable long-context processing capabilities (OpenAI, 2024; Yang et al., 2024; et al., 2024). Despite their effectiveness, LLMs encounter *efficiency challenges* in processing long contexts. On one hand, transformer-based LLMs incur substantial computational costs due to the quadratic complexity of self attention. On the other hand, they require tremendous GPU memory to hold the KV cache of the entire sequence for faster decoding. Both computation and memory costs increase as the context length grows.

A wide array of studies are dedicated to alleviating efficiency issues, among which context compression is a promising direction (Mu et al., 2023; Chevalier et al., 2023; Ge et al., 2024; Jiang et al., 2023a;b). This approach aims to compress raw input into more concise representations, allowing the generation process to be conditioned on a shorter context. Therefore, it helps to reduce both computation cost of inference and memory cost from KV cache, while also enabling the processing of longer inputs than the LLM's built-in context window.

Despite the current progresses, it it remains a tough challenge to compress long contexts. Specifically, existing methods usually summarize the context into a few soft tokens (Chevalier et al., 2023; Ge et al., 2024), which constitute the major bottleneck to summarize the complex information within

---

[*]Peitian Zhang and Zheng Liu are the co-first authors
[†]Zheng Liu is the corresponding author

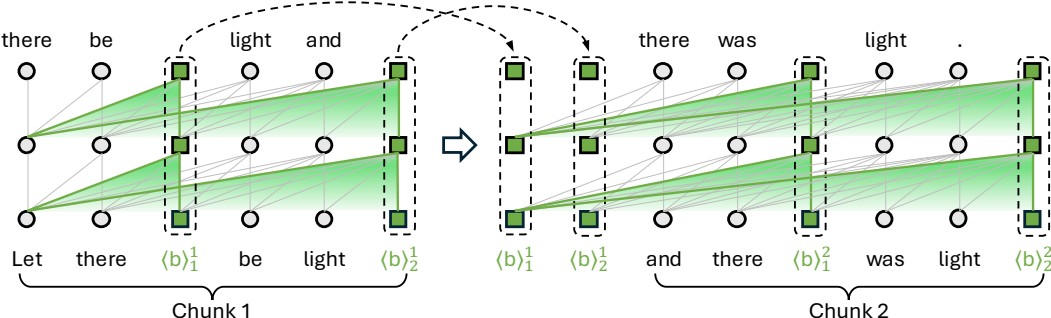

Figure 1: Overview of Activation Beacon. The context is partitioned into chunks. Each chunk is further split into fine-grained units and interleaved with beacon tokens according to a compression ratio (2 in the figure). The LLM encodes one chunk at a time, compressing the context into beacon tokens' activations, which are *accumulated* and *reused* for encoding following chunks.

long contexts. Besides, they try to compress the context "all-at-once", lacking a fine-grained handling of the detailed information. Moreover, these soft tokens must be re-encoded before generation, resulting in inferior efficiency in both training and inference. Lastly, these methods are learned to compress with a fixed number of soft tokens, thus, it's hard to customize the compression ratio for downstream tasks. While some alternamtive methods focus on deleting unimportant tokens (Jiang et al., 2023b; Li et al., 2024b), they depend on the input question to estimate the token importance, limiting their efficiency in real-world multi-turn scenarios.

To address the above challenges, we present Activation Beacon (Figure 1), a plug-in module to transformer-based LLMs that enables effective, efficient, and flexible compression of long contexts. Activation Beacon is featured with the following technical designs.

First of all, we introduce a new special token, called the beacon token $\langle b \rangle$. The context is distilled into beacon tokens' *activations* (i.e. keys and values at every layer), whose capacity are large enough to encapsulate the complex information within long contexts.

Next, we tailor the compression workflow, where each fine-grained context unit is progressively compressed. Specifically, the long context is partitioned into equal-size chunks. Each chunk is further split into fine-grained units of size $\alpha$ where $\alpha$ is the desired compression ratio. A group of beacon tokens are interleaved with these units (one beacon token is dispatched to the end of every unit). The LLM encodes one chunk at a time, distilling the chunk's information into beacon tokens' activations during self attention. After encoding, the raw tokens' activations are *discarded*; while the beacon tokens' activations are *accumulated* and *reused* for encoding following chunks. This progressive workflow brings forth several advantages: 1) It can handle inputs longer than the backbone LLM's context window as the chunk size is small. 2) It achieves fine-grained compression since the attention scope of each beacon token is differentiated. 3) By caching and reusing activations, it facilitates contiguous gradient propagation in training, avoids re-encoding overhead in inference, and allows for incrementally updating the compression results in multi-turn scenarios.

Finally, Activation Beacon is learned with compression-based auto-regression to optimize the generation quality conditioned on the compressed context. Thanks to high sample efficiency, the model can be effectively trained with 1B plain corpus and 30K fine-tuning samples (maximum context length is 20K), which can be quickly accomplished. During training, we randomly sample the compression ratio for each chunk, enhancing the model's flexibility to tackle different compression ratios in downstream tasks. Note that all beacon tokens share the same token embedding, one can use arbitrary number of beacon tokens to achieve the desired compression ratio by repeating.

In our experiments, Activation Beacon is applied to Llama-2 (Touvron et al., 2023) and Qwen-2 (Yang et al., 2024). We evaluate the resulted models on a variety of long-context tasks (whose lengths may be much longer than the training length, e.g., 128K), such as document understanding, few-shot learning, and Needle-in-a-Haystack. Whilst existing methods struggle to handle these challenging tasks, Activation Beacon maintains a comparable performance to the uncompressed baseline across various compression configurations, meanwhile achieving 2x acceleration and 8x KV cache reduction. Moreover, the LLM's original capabilities on short context is well preserved.

## 2 RELATED WORKS

Recently, processing long context has become a fundamental capability of modern LLMs (OpenAI, 2024; et al., 2024; Yang et al., 2024; DeepSeek-AI, 2024). The recipe of context window extension is roughly the same: modifying the rotary position embedding (Su et al., 2021) by extrapolation and interpolation (Chen et al., 2023a; ntk, 2023; Peng et al., 2023; Ding et al., 2024), and leveraging long-dependency data in both the pre-training and post-training stage. Despite the impressive progress in effectiveness, LLMs face significant challenges in efficiency. There is significant computational cost due to the quadratic complexity of transformer, and huge memory cost because LLMs need to hold the KV activations of the entire sequence on GPU for faster decoding. Multiple threads of research endeavour to reduce these costs, which are discussed as follows.

**Sparse Attention.** Conventional sparse attention methods require re-training a model from scratch using the designated sparse patterns (Zaheer et al., 2020; Beltagy et al., 2020). However, extensive recent studies have identified that the attention pattern of LLMs are naturally sparse despite they are densely trained (Jiang et al., 2024a; Xiao et al., 2023; Han et al., 2023; Zhu et al., 2024). They also propose to dynamically set appropriate sparse patterns for each head so that the attention mass can be largely preserved, leading to competitive performance against the full-attention method with reduced computation. However, these methods require holding all KV activations on chip to dynamically determine the optimal sparse patterns, making them unsuitable for KV cache reduction. There are some sparse attention methods that directly evict the middle tokens (Han et al., 2023; Xiao et al., 2023). Despite their high efficiency and ability to generate endless fluent texts, these methods' cannot memorize information in the middle contexts, leading to inferior performances on long-context tasks Xiao et al. (2024).

**KV Compression.** This line of research focuses on compressing the KV activations to reduce the attention computation as well as the cache size. Since the KV activations are per-layer, per-head, per-token, and per-channel float numbers, they can be reduced from all the five dimensions (including the numerical dimension). For example, CLA (Brandon et al., 2024) shares the KV cache across multiple layers; GQA (Ainslie et al., 2023) compresses multiple key/value heads into a single one; MLA (DeepSeek-AI, 2024) compresses the channels into fewer and more compact ones; and KIVI (Zirui Liu et al., 2023) quantizes the numerical value in the activations. The sequence-wise compression (also known as context compression), where Activation Beacon falls, is introduced in the following paragraph. It is orthogonal to the compression along other dimensions, and the complementary effect of the compression along different dimension could be left for future work. Besides, some recent studies design efficient strategies for offloading and transferring KV cache (Liu et al., 2023; Xiao et al., 2024). They can also be jointly used with KV compression techniques to achieve more efficient long-context generation.

**Context Compression.** This type of methods aim to compress the raw context into shorter yet more compact representations. Existing studies are usually tailored for compressing short context (less than 1K), which tend to be sub-optimal for long-context compression. Specifically, Gisting (Mu et al., 2023) compresses the user instruction into gist activations all at once. As a result, it cannot process context longer than the backbone LLM's window. CCM (Kim et al., 2024) extends Gisting to compress conversations in online chatting, yet it cannot be used in general long context tasks such as long document understanding. ICAE (Ge et al., 2024) and AutoCompressor (Chevalier et al., 2023) alleviate this problem by segmenting the long context into chunks and compressing each chunk, in order to compress contexts longer than the backbone LLM's window. CEPE (Yen et al., 2024) shares a similar workflow while introducing a standalone encoder to compress the context and utilizing the compression results through a cross-attention module. However, these methods compress the context into soft tokens, which are the major bottleneck to encapsulate the complex information in long contexts. Their compression workflow also lacks fine-grained handling of the chunked inputs, resulting in inferior compression quality. Moreover, these methods must perform re-encoding or employ additional cross-attention mechanism to utilize the compressed soft tokens, which introduces extra overhead. Lastly, since the number of soft tokens are pre-defined, it is hard to flexibly assign the compression ratio for downstream tasks. Another branch of methods (Jiang et al., 2023b; Li et al., 2024b) propose to delete unimportant tokens to realize compression. However, they depend on the input question to accurately estimate the token importance, leading to low efficiency in real-world multi-turn scenarios. Compared with existing approaches, Activation Beacon is able to achieve more effective, efficient, and flexible compression. Based on context compression techniques, there are

some innovated frameworks like LLoCO (Tan et al., 2024). It is built upon a compressor and a decoder, where the context is compressed offline and offloaded into a retrieval system. The decoder then efficiently responds to the user inputs based on retrieved compression results. Both modules are learned with in-domain fine-tuning. Our work aims at improving the compressor itself, and hence is orthogonal to these frameworking research.

## 3 METHODOLOGY

LLMs accomplish arbitrary tasks in the form of next-token prediction. Formally, given the context $X = [x_1, \ldots, x_n]$, the LLM generates the next token based on all preceding tokens and its well-trained parameters: $\Pr(x_{n+1} \mid x_1, \ldots, x_n; \boldsymbol{\Theta})$. Transformer-based LLMs incur heavy computation cost due to the quadratic complexity of self attention; besides, they require tremendous GPU memory to store the KV cache of $x_{\leq n+1}$ for faster decoding (Zhang et al., 2023). Both the costs in computation and memory significantly expand when the context length increases.

Activation Beacon employs a new special token, namely beacon token $\langle \text{b} \rangle$, and condenses the raw context $X$ into beacon tokens' activations $\boldsymbol{\Psi}$ (i.e. their keys and values at every layer). The next-token prediction is converted to condition on the compressed context instead of the plain one. Given $|\boldsymbol{\Psi}| < |X|$, both the computation cost and the KV cache size are reduced. Additionally, the LLM is enabled to handle context longer than its window size based on the compressed representations. We tailor the compression mechanism and the learning method of Activation Beacon towards achieving effective, efficient, and flexible compression, which will be elaborated in the following.

### 3.1 COMPRESSION MECHANISM

**Overview.** We propose to *progressively compress each fine-grained units of long contexs.* Specifically, given the input context $X$ whose length may exceed the LLM's context window $N$, it is first partitioned into chunks of the same size $w$ (e.g., 1024):

$$[x_1, \ldots, x_n] \xrightarrow{\text{Partition}} [X_1, \ldots X_{\lceil n/w \rceil}], \ X_i = [x_{(i-1)w+1}, \ldots, x_{iw}]^1 = [x_1^i, \ldots, x_w^i]. \quad (1)$$

Next, for each chunk $X_i$, we determine a compression ratio $\alpha_i$ ($w$ is evenly divisible by $\alpha_i$). The chunk is further split into fine-grained units of size $\alpha$. Then a group of $k_i = w/\alpha_i$ beacon tokens, $B_i = [\langle \text{b} \rangle_1^i, \ldots, \langle \text{b} \rangle_{k_i}^i]$, are *interleaved* with these units. In other words, one beacon token is dispatched to the end of every unit:

$$X_i \xrightarrow{\text{Interleave } B_i} X_i' = [x_1^i, \ldots, x_{\alpha_i}^i, \langle \text{b} \rangle_1^i, \ \ldots \ , x_{w-\alpha_i+1}^i, \ldots, x_w^i, \langle \text{b} \rangle_{k_i}^i]. \quad (2)$$

The LLM encodes these chunks *one by one*, compressing the contextual information of each chunk into the corresponding beacon tokens' activations during self attention. After encoding $X_i'$, we *discard* activations of all the raw tokens $X_i$, while we *accumulate* the activations of the beacon tokens $B_i$. When encoding the next chunk $X_{i+1}'$, the LLM directly conditions on the accumulated beacon activations as a proxy to the raw context $X_{\leq i}$.

This progressive workflow benefits both compression quality and running efficiency. On one hand, it enables thorough distillation of complex information within long contexts and allows for the compression of inputs that exceed the LLM's context window. On the other hand, by caching and reusing beacon tokens' activations, it avoids redudant computation and allows for incrementally update of the compression results in multi-turn interactions.

**Encoding and Compression.** As shown in Figure 2, Activation Beacon reuses all modules of the LLM except imposing a slight modification on self attention. Without loss of generality, for the $i$-th chunk $X_i'$, the encoding process can be written as:

$$\text{LLM}( \ \underbrace{\langle \text{b} \rangle_1^i, \ldots, \langle \text{b} \rangle_{k_{i-1}}^{i-1},}_{\text{beacon activations accumulated from } X_{<i}'} \ \underbrace{x_1^i, \ldots, x_{\alpha_i}^i, \langle \text{b} \rangle_1^i, \ \ldots \ , x_{w-\alpha_i+1}^i, \ldots, x_w^i, \langle \text{b} \rangle_{k_i}^i}_{\text{the current chunk } X_i'} ), \quad (3)$$

where the input to the LLM is a mix of the activations accumulated from previous chunks and the tokens to be encoded within the current chunk. Let $D$ denote the LLM's hidden size, $\boldsymbol{H} \in$

---

[1]The last chunk $X_{\lceil t/w \rceil}$ may be shorter than $w$, which is omitted for simplicity.

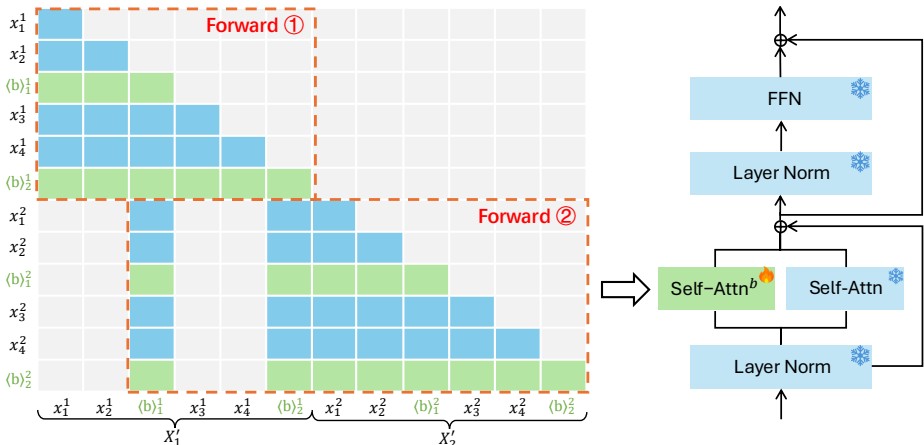

Figure 2: Activation Beacon performs compression during self attention while reusing all other modules of the LLM. *Forward* ①: encode and compress the first chunk. *Forward* ②: encode and compress the second chunk conditioned on activations of preceding beacon tokens.

$\mathbb{R}^{(w+k_i)\times D}$ denote input hidden states to self attention in an arbitrary layer of the LLM. We first slice out the hidden states of raw tokens and beacon tokens:

$$\mathbb{I}^r = \{j \mid x_j^i \neq \langle b\rangle\}, \quad \mathbb{I}^b = \{j \mid x_j^i = \langle b\rangle\}; \quad \boldsymbol{H}^r = \boldsymbol{H}[\mathbb{I}^r], \quad \boldsymbol{H}^b = \boldsymbol{H}[\mathbb{I}^b]. \tag{4}$$

Then the hidden states are projected into queries, keys, and values:

$$\boldsymbol{Q}^r = \boldsymbol{W}_Q^r \boldsymbol{H}^r, \qquad \boldsymbol{K}^r = \boldsymbol{W}_K^r \boldsymbol{H}^r, \qquad \boldsymbol{V}^r = \boldsymbol{W}_V^r \boldsymbol{H}^r,$$
$$\boldsymbol{Q}^b = \boldsymbol{W}_Q^b \boldsymbol{H}^b, \qquad \boldsymbol{K}^b = \boldsymbol{W}_K^b \boldsymbol{H}^b, \qquad \boldsymbol{V}^b = \boldsymbol{W}_V^b \boldsymbol{H}^b, \tag{5}$$

where $\boldsymbol{W}_*^r$ are the LLM's original projection matrices and $\boldsymbol{W}_*^b$ are the newly introduced matrices to handle beacon tokens only. Afterwards, the query/key/value states of raw tokens and beacon tokens are scattered back to acquire $\boldsymbol{Q}, \boldsymbol{K}, \boldsymbol{V} \in \mathbb{R}^{(w+k_i)\times D}$:

$$\boldsymbol{Q}[\mathbb{I}^r] = \boldsymbol{Q}^r,\ \boldsymbol{Q}[\mathbb{I}^b] = \boldsymbol{Q}^b; \quad \boldsymbol{K}[\mathbb{I}^r] = \boldsymbol{K}^r,\ \boldsymbol{K}[\mathbb{I}^b] = \boldsymbol{K}^b; \quad \boldsymbol{V}[\mathbb{I}^r] = \boldsymbol{V}^r,\ \boldsymbol{V}[\mathbb{I}^b] = \boldsymbol{V}^b. \tag{6}$$

Finally, the standard self-attention is computed over the entire input:

$$\boldsymbol{A} = \mathrm{softmax}\left(\mathrm{mask}\left(\frac{\boldsymbol{Q}\left\{\boldsymbol{K}^{ac}; \boldsymbol{K}\right\}^T}{\sqrt{D}}\right)\right), \quad \boldsymbol{V} = \boldsymbol{A}\left\{\boldsymbol{V}^{ac}; \boldsymbol{V}\right\}. \tag{7}$$

In the above equations, $\{\cdot\,;\,\cdot\}$ denotes matrix concatenation. $\boldsymbol{K}^{ac}, \boldsymbol{V}^{ac} \in \mathbb{R}^{m_{i-1}\times D}$ are the beacon tokens' activations accumulated from previous chunks where $m_{i-1} = \sum_{j=1}^{i-1} k_j$, and $\mathrm{mask}$ denotes the causal attention mask. During self attention, all tokens are encoded by their relative positions ($[m_{i-1}, \ldots, m_i + w - 1]$ for queries and $[0, \ldots, m_i + w - 1]$ for keys). The value states $\boldsymbol{V}$, are further processed by other modules (e.g., output projection, MLP, and LayerNorm) before passing to the next layer. After self attention, the keys and values of beacon tokens, i.e. $\boldsymbol{K}^b$ and $\boldsymbol{V}^b$, have distilled the contextual information of $X_i$. They are incrementally accumulated:

$$\boldsymbol{K}^{ac} = \{\boldsymbol{K}^{ac}; \boldsymbol{K}^b\}, \quad \boldsymbol{V}^{ac} = \{\boldsymbol{V}^{ac}; \boldsymbol{V}^b\}. \tag{8}$$

In our default setting, the beacon tokens are interleaved with raw tokens. This leads to a differentiated attention scope for each beacon token ($\langle b\rangle_j^i$ attends to one more interval than $\langle b\rangle_{j-1}^i$), contributing to the *fine-grained* compression of the context. We also explore the setting to dispatch all beacon tokens at the end of the chunk, which results in inferior compression quality (§4.6).

Note that unlike ICAE (Ge et al., 2024) and LLMLingua (Jiang et al., 2023b), Activation Beacon unifies generation and compression operations within a single forward pass of the LLM. That is to say, the hidden states of the last input token $\boldsymbol{H}[\mathbb{R}^r[-1]]$ is directly used to decode the next token without resorting to another decoder model.

**Efficiency Analysis.** Activation Beacon reduces the KV cache by $\alpha$ times where $\alpha$ is the *average compression ratio* and hence the memory cost. This is because it only needs to store the compressed

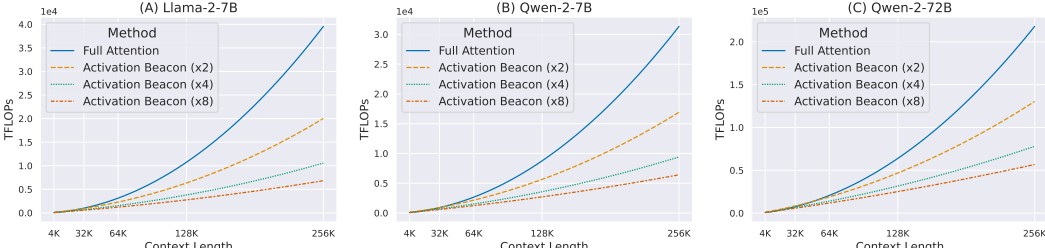

Figure 3: Comparison of the forward FLOPs of different models using full attention and Activation Beacon (the compression ratio is annotated in the brackets).

activations of the preceding chunks instead of the raw activations. In terms of computation, the situation is a bit more complex. Specifically, Activation Beacon significantly reduces the computation in self attention, because each token only needs to interact with local tokens within the chunk and preceding beacon tokens, which are approximately $\alpha$ times shorter than the raw context. However, it also triggers more computation to encode the inserted beacon tokens in other modules (e.g., MLP). Formally, given an LLM with a fixed number of layers, attention heads, and hidden size, let $s$ denote the input context length, $s^{pst}$ denote the cached context length, the forward FLOPs is:

$$\text{FLOPs} = F^{Att}(s, s^{pst}) + F^{Oth}(s), \tag{9}$$

where $F^{Att}$ is the computation during self attention, and $F^{Oth}$ is the computation of other modules. For full-attention models, $s = n, s^{pst} = 0$. For "beaconed" models, the FLOPs is:

$$\text{FLOPs}^{bcn} = \sum_{i=1}^{\lceil \frac{n}{w} \rceil} F^{Att}\left(\frac{(\alpha+1)w}{\alpha}, \frac{(i-1)w}{\alpha}\right) + F^{Oth}(n + \lceil \frac{n}{\alpha} \rceil). \tag{10}$$

Since the implementation of $F^{Att}$ and $F^{Oth}$ depends on the actual setting of the LLM (see Appendix B), we visualize the FLOPs curve of three different LLMs in Figure 3. It can be observed that Activation Beacon consistently saves computational costs across different model settings and scales. The extent of saving amplifies as the context length grows, finally achieving more than x4 reduction at 256K context. The specific implication on latency is studied in §4.3.

## 3.2 LEARNING METHOD

**Compression-Based Auto-Regression.** Activation Beacon is learned to optimize the generation quality conditioned on the mixture of the compressed context and the local context. Formally, the compression-based next-token prediction loss is minimized:

$$\min_{\Theta^b}. \sum_{i=2}^{\lceil N/w \rceil} \sum_{j=1}^{w} \text{Pr}(x_j^i \mid \langle \text{b} \rangle_1^1, \dots, \langle \text{b} \rangle_{k_{i-1}}^{i-1}, x_1^i, \dots x_{j-1}^i; \Theta, \Theta^b). \tag{11}$$

$\Theta$ denotes the parameters of the LLM itself, which are *fixed* throughout the training process. $\Theta^b$ includes the projection matrices for beacon tokens at each layer $\boldsymbol{W}_Q^b, \boldsymbol{W}_K^b, \boldsymbol{W}_V^b$, and the token embedding of beacon token $\boldsymbol{e}_{\langle \text{b} \rangle}$ (we use one *shared* embedding for all beacon tokens). The training loss can be obtained from all tokens except the ones in the first chunk. Such a property leads to high sample efficiency that maximizes the use of training data. Note that we exclude the beacon tokens from the above loss (setting their labels to -100) because they are solely intended for compression.

**No Stop Gradients.** Recurrent memory methods (Chevalier et al., 2023; Bulatov et al., 2023) stop the gradients back-propagation at a given chunk number to improve the training efficiency. This is because these methods depend on the *final-layer* outputs of preceding chunks to encode the current chunk, which results in deepened computation graph as more chunks are involved. In contrast, Activation Beacon only depends on the *previous-layer* outputs of preceding chunks (the encoding of $X_i'$ at layer $l$ only conditions on the results of $X_{i-1}'$ at layer $l-1$), which is the same as any auto-regressive LLMs. Thus, the gradients can naturally flow through all chunks to optimize the compression effect over long contexts.

**Chunk-Wise Random Compression Ratio.** To teach the model to flexibly support diverse compression granularities, the compression ratio $\alpha_i$ for the $i$-th chunk is *randomly sampled* from

Table 1: Evaluation on LongBench (Bai et al., 2023). Activation Beacon maintains comparable performance to the uncompressed baseline (Full-FT), outperforming other compression methods. "Length" indicates the number of tokens in the *input* context.

| Model | Method | Length | Single-Doc | Multi-Doc | Summ. | Few-Shot | Code |
|-------|--------|--------|------------|-----------|-------|----------|------|
| Llama-2-7B | Full | 4K | 24.7 | 22.4 | 24.6 | **63.2** | 57.7 |
| | Full-FT | 32K | 34.8 | **27.5** | 23.2 | 61.8 | **57.8** |
| | AutoCompr. | 32K | 12.9 | 16.4 | 16.3 | 23.8 | 39.4 |
| | ICAE | 32K | 19.5 | 19.2 | 19.5 | 24.8 | 27.8 |
| | LongLLML. | 32K | 21.5 | 18.8 | 21.7 | 49.5 | 53.2 |
| | SnapKV | 4K | 24.2 | 22.6 | 16.3 | 60.1 | 57.7 |
| | Ours | 32K | **34.9** | **27.5** | **25.0** | 61.4 | **57.8** |
| Qwen-2-7B | Full | 32K | 38.8 | 37.5 | 26.7 | **70.1** | 60.3 |
| | Full-FT | 32K | **41.0** | **40.6** | 26.8 | 68.5 | 66.1 |
| | LongLLML. | 32K | 24.7 | 20.3 | 26.3 | 55.9 | 50.1 |
| | SnapKV | 32K | 38.7 | 37.6 | 26.2 | 67.1 | 60.3 |
| | Ours | 32K | 40.5 | 40.3 | **26.8** | 68.4 | **66.4** |

$\{2, 4, 8, 16, 32\}$ during training. At inference, one can choose one compression ratio according to the specific efficiency requirement in downstream tasks and stick to it for all chunks.

## 4 EXPERIMENTS

Our experiment mainly study Activation Beacon's effectiveness (§4.2), efficiency (§4.3), and flexibility (§4.4) in long context compression. Besides, we explore Activation Beacon's impact on short-context capabilities of the backbone LLM (§4.5) and the effect of each technical design (§4.6).

### 4.1 SETTINGS

**Implementation.** Activation Beacon is applied to Llama-2-7B (chat)[2] and Qwen-2-7B (instruct). The chunk size $w$ is 1024 for Llama-2 and 2048 for Qwen-2. FlashAttention-2 (Dao, 2023) is used to speed up attention computation. For all our experiments, we use Huggingface framework (Wolf et al., 2020) and one 8xA800 (80G) machine.

**Training.** The training consists of two phases. In pre-training, we use 1B tokens sampled from Red-Pajama (Computer, 2023). The eos token is appended to the end of every document. In fine-tuning, we leverage LongAlpaca (Chen et al., 2023b), BookSum (Kryściński et al., 2022), and synthetic data from GPT-3.5 (details in Appendix A). All the training samples are shorter than 20K. The batch size is 8. The learning rate is 5e-5 for pre-training and 1e-5 for fine-tuning, with linear decay and no warmup. As introduced, the LLM's original parameters are frozen throughout the training process.

**Baselines.** We compare Activation Beacon with the uncompressed baseline (denoted as Full) and the uncompressed baseline fine-tuned with the same training data (denoted as Full-FT). Besides, we include the following context compression methods that can tackle long context for comparison, including AutoCompressors (Chevalier et al., 2023), ICAE (Ge et al., 2024), LongLLMLingua (Jiang et al., 2023b), and SnapKV (Li et al., 2024b). The first two methods only support Llama-2. To guarantee fair comparison, we fine-tune their official checkpoints using the same training data.

### 4.2 COMPRESSION EFFECTIVENESS

To verify the compression effectiveness of Activation Beacon, we evaluate it on LongBench (Bai et al., 2023), which consists of a variety of long-context tasks with 32K maximum length, including question answering, summarization, few-shot learning, and code completion. Since Llama-2 has a context window of 4K, we truncate the context longer than 4K from middle before inputting to it. For compression methods implemented on Llama-2, we set *adaptive* compression ratio, translating to x2 compression for 4K-8K contexts, x4 compression for 8K-16K contexts, and x8 compression for 16K-32K contexts. For methods implemented on Qwen-2, we apply a uniform compression ratio of x4. The results are reported in Table 1. We highlight two observations in the following.

---

[2]We use Llama-2 because AutoCompressor and ICAE are based on it, both of which are important baselines.

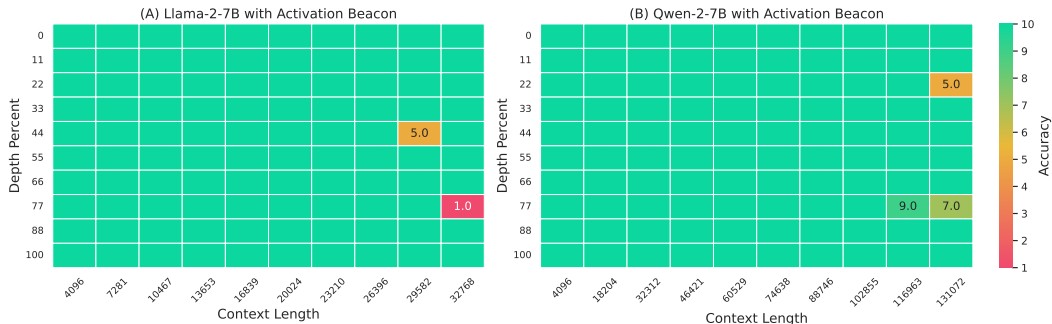

Figure 4: Evaluation on Needle-in-a-Haystack. Activation Beacon can accurately retrieves the needle most of the time, despite the context is far longer than its training data.

Firstly, **Activation Beacon achieves superior compression quality over other compression baselines across all tasks.** Concretely, it siginificantly outperforms ICAE and AutoCompressor, which verifies that several soft tokens are not enough to encapsulate the rich information within long contexts. LongLLMLingua also lags far behind Activation Beacon because it need to delete too many tokens given a high compression ratio (e.g., x4, x8), which may destroy the coherence of the context and lose important information. Despite SnapKV's top performance among baselines, it cannot compress context longer than the backbone LLM's window. This is because it estimates the token importance based on self attention, which becomes inaccurate once the context exceeds the window size, limiting its practical usage when compressing long contexts.

Secondly, **Activation Beacon achieves comparable performance to the fine-tuned uncompressed baseline (Full-FT)** even though Full-FT takes in the entire context without compression. This indicates that Activation Beacon is able to compress long contexts without evident information loss, which validates its high compression quality yielded from the progressive compression workflow. Furthermore, Activation Beacon improves upon Llama-2 by a large margin despite their context window is the same, i.e. 4K. The gain is because Llama-2 (Full) directly uses the truncated 4K context, while Activation Beacon compresses the 32K context into 4K compact activations. This implies that Activation Beacon can effectively introduce useful information from Llama-2's unseen context. Therefore, it can be viewed as an efficient approach for context extension.

We further evaluate Activation Beacon on Needle-in-a-Haystack (NIAH) following the official settings (gkamradt, 2023) to investigate whether it will lose fine-grained information. The accuracy is estimated by ChatGPT (ranges from 1 to 10). For both Llama-2 and Qwen-2, we set adaptive compression ratio as introduced above. The results are shown in Figure 4. It can be observed that Activation Beacon **precisely retrieves the needle** most of the time. Note that Activation Beacon conducts *query-independent* compression, which means it has no prior knowledge of what to compress and what not. Hence, this remarkable performance again validates our tailored compression mechanism and learning method can preserve the fine-grained contextual information. Moreover, Activation Beacon is only trained on context shorter than 20K, while its compression capability can generalize to far longer contexts (e.g., 128K).

## 4.3 COMPRESSION EFFICIENCY

We evaluate the efficiency of Activation Beacon based on the Multi-Needle-in-a-Haystack task following NeedleBench (Li et al., 2024a). Specifically, we fix the context length to 32K for Llama-2 and 128K for Qwen-2, and insert 3 different needles at different positions. The task is organized in a multi-turn conversation setting, where the model is asked to retrieve one specific needle in each turn. The experiment is repeated 20 times for each model with distinct needle positions. In Table 2, we report the accuracy and the end-to-end latency of compression & generation (measured in seconds).

It can be observed that **Activation Beacon enjoys lower latency than other compression baselines**. Notably, it is 1.8x faster than AutoCompressor because it does not have to re-encode the soft tokens from previous chunks. It also leads to 9.3x and 3.6x acceleration upon LongLLMLingua and SnapKV given three turns, respectively. This is because both baselines are query-dependent while Activation Beacon is not, which eliminates the need to re-compute the compression results for

Table 2: Evaluation on Multi-Needle-in-a-Haystack where the questions are issued one-by-one in a multi-turn conversation setting. All compression methods use a x8 compression ratio. Activation Beacon consistently outperforms other compression baselines while enjoying lower latency, especially when the context lengthens and the turn number increases.

| Model | Length | Method | 1-Turn Acc | 1-Turn Latency | 2-Turn Acc | 2-Turn Latency | 3-Turn Acc | 3-Turn Latency |
|---|---|---|---|---|---|---|---|---|
| Llama-2-7B | 32K | Full-FT | **9.75** | 1.336 | **9.45** | 1.532 | **9.10** | 1.726 |
| | | AutoCompr. | 1.60 | 2.135 | 1.50 | 2.561 | 1.50 | 2.994 |
| | | ICAE | 2.15 | 1.182 | 2.15 | 1.476 | 2.00 | 1.805 |
| | | LongLLML. | 2.05 | 2.813 | 2.00 | 5.062 | 2.00 | 7.034 |
| | | SnapKV | 1.00 | **0.859** | 1.00 | 1.656 | 1.00 | 2.199 |
| | | Ours | **9.75** | 1.153 | 9.40 | **1.356** | 9.05 | **1.638** |
| Qwen-2-7B | 128K | Full-FT | **9.75** | 4.399 | **9.50** | 5.254 | **9.20** | 6.153 |
| | | LongLLML. | 2.00 | 10.455 | 1.55 | 19.768 | 1.50 | 27.751 |
| | | SnapKV | 9.45 | 3.955 | 8.95 | 7.803 | 8.85 | 10.659 |
| | | Ours | 9.70 | **2.445** | 9.35 | **2.773** | 9.10 | **2.981** |

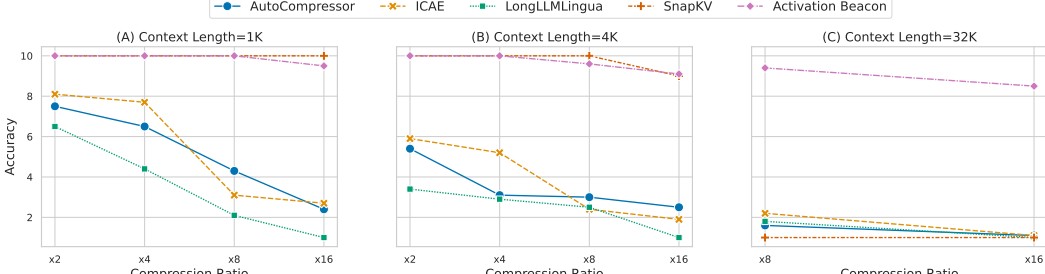

Figure 5: Evaluation on Needle-in-a-Haystack with various compression ratios based on Llama-2. Activation Beacon achieves top compression quality across all compression configurations.

different input questions. Moreover, Activation Beacon demonstrates consistent speed-up over the Full-FT baseline, achieving **2x acceleration at 128K context length**. This matches our estimation in Figure 3(b) as Activation Beacon (x8) saves half of the computation. In the meanwhile, since the compression ratio is x8, it leads to **8x reduction of the KV cache**. Lastly, Activation Beacon always attains nearly-lossless generation quality against the uncompressed baseline, which is in line with previous observations.

## 4.4 COMPRESSION FLEXIBILITY

Activation Beacon is learned to support various compression ratios during training. In Figure 5, we evaluate its compression quality under different compression ratios and context lengths. According to the figure, Activation Beacon maintains top accuracy across all compression ratios, outperforming most compression baselines by a large margin. Though SnapKV performs on par with our method at 1K and 4K context length, it fails to compress inputs longer than the LLM's window size, which may limit its practical usage. To summarize, Activation Beacon is a flexible solution to long context compression with the support of diverse compression ratios and various context lengths. Generally, we recommend to use x8 compression ratio as it preserves most information with high efficiency.

## 4.5 SHORT-CONTEXT CAPABILITIES

Since Activation Beacon interleaves beacon tokens with raw tokens and is primarily trained with long-context tasks, it is intriguing to examine whether the current recipe will impair the short-context capabilities of the backbone LLM.

Table 3: Activation Beacon preserves the short-context capabilities of the backbone LLM.

| Model | Method | MMLU | ARC-C | BoolQ | GSM8K |
|---|---|---|---|---|---|
| Llama-2-7B | Full | 47.5 | 48.5 | 86.2 | 9.2 |
| | Ours | 46.6 | 48.4 | 86.5 | 9.3 |
| Qwen-2-7B | Full | 70.1 | 62.7 | 87.1 | 76.0 |
| | Ours | 69.1 | 62.7 | 87.2 | 76.2 |

In Table 3, we compare Activation Beacon with the original LLM (Full) on popular benchmarks, including MMLU (Hendrycks et al., 2021), ARC-Challenge (Bhakthavatsalam et al., 2021), BoolQ (Clark et al., 2019), and GSM8K (Cobbe et al., 2021). We can observe that Activation Beacon leads to very little performance degradation on short-context tasks. In other words, the short-context capabilities are well preserved. We conjecture that the primary reason is the LLM's original parameters are frozen throughout the training process.

## 4.6 Ablation Studies

We study the impact of each technical factor, including the compression of fine-grained context units, the sampling strategy of compression ratio, and training stages. The experiments are based on Qwen-2-7B and Single-Doc QA task from LongBench (32K context with x4 compression ratio). The results are shown in Table 4. Firstly, instead of splitting the chunk into fine-grained units and interleaving beacon tokens, we append all beacon tokens at the end of the chunk so that their attention scopes are the same. It can be observed that such operation results in significant information loss after compression, which justifies the effectiveness of our fine-grained compression mechanism. Secondly, we replace the chunk-wise random compression ratio with the instance-wise one, which randomly selects one compression ratio for each training instance rather than each chunk. We can observe that the chunk-wise setting facilitates better learning of the compression functionality. Lastly, we remove either pre-training or fine-tuning. It can be observe that both stages are useful, and the combination of both leads to the optimal performance. This also implies that the compression quality of Activation Beacon can be further enhanced given more abundant and targeted training.

Table 4: The impact of different technical factors.

| Method | Single-Doc |
|---|---|
| Default | **40.5** |
| w/o Fine-Grained Compression | 35.2 |
| w/o Chunk-Wise Random Ratio | 37.7 |
| w/o Pre-training | 34.9 |
| w/o Fine-tuning | 35.5 |

## 5 Conclusion

This paper introduces Activation Beacon, a plug-in for transformer-based LLMs to enable effective, efficient, and flexible compression of long contexts. Activation Beacon is featured for several critical innovations, including the progressive and fine-grained compression workflow to distill the context into a small set of activations, the compression-based auto-regression to optimize the model with high sample efficiency, and the random sampling of compression ratios to support various downstream scenarios. According to extensive experimental evaluations, Activation Beacon consistently outperforms existing context compression methods across various compression configurations. It even maintains comparable performance to the uncompressed baseline, meanwhile achieving 2x acceleration and 8x KV cache reduction. Moreover, the short-context capabilities of the LLM is well preserved.

## 6 Acknowledgements

This work is supported by the National Science and Technology Major Project (2023ZD0121504), National Natural Science Foundation of China No. 62272467, and Beijing Municipal Science and Technology Project No. Z231100010323009.

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

## A  TRAINING DATA

In the pre-training phase, we use 1B tokens from RedPajama. We add an eos token to the end of each document. The documents longer than 20480 or shorter than 2400 are removed.

---

**Prompt A.1**

{Segment 1}
...
{Segment N}

Question: {Question 1 for Segment 1}
Answer: {Answer 1 for Segment 1}
...
Question: {Question 4 for Segment N}
Answer: {Answer 4 for Segment N}

---

In the fine-tuning phase, we use three datasets. 1) LongAlpaca (Chen et al., 2023b), which contains long-context QA and summarization data; 2) BookSum (Kryściński et al., 2022), which contains chapter-level summarization of books. 3) Synthesized QA dataset. This dataset contains 16K long-context question answering instances (13K for books and 3K for papers). Specifically, we split a

Table 5: Comparison of the breakdown latency of prefilling and decoding, as well as peak GPU memory between the full-attention baseline and Activation Beacon. Activation Beacon uses x8 compression uniformly.

| Model | Input Length | Method | Prefilling | Decoding | Total | GPU Memory |
|---|---|---|---|---|---|---|
| Qwen-2.5-7B | 32K | Full | 0.522 | 0.709 | 1.231 | 27.0 |
| | | Ours | 0.514 | 0.506 | 1.020 | 18.1 |
| | 128K | Full | 3.312 | 2.081 | 5.393 | 64.3 |
| | | Ours | 2.038 | 0.550 | 2.588 | 24.6 |
| Qwen-2.5-14B | 32K | Full | 0.952 | 1.862 | 2.814 | 45.7 |
| | | Ours | 0.996 | 0.857 | 1.853 | 36.0 |
| | 128K | Full | – | – | – | OOM |
| | | Ours | 4.340 | 1.117 | 5.457 | 40.3 |

given long context (a paper or a book) into short segments (a chunk with less than 4096 tokens) using the SemanticTextSplitter[3]. For each segment, we prompt GPT-3.5-turbo to generate 4 question-answer pairs based on the segment. We then group continuous segments using Template A.1, where we can control the resulted context length by concatenating different number of segments. The books are randomly sampled from Books3 corpus, and the papers are sampled from Arxiv, both coming from the Pile (Gao et al., 2020). All the above fine-tuning data are formatted in the manner of multi-turn conversations and limit the context length up to 20480. To mitigate forgetting, we also include 5000 samples from the pre-training data.

## B FLOPs CALCULATION

Denote the input sequence length as $s$, the cached sequence length as $s^{pst}$, query head number as $h^q$, key/value head number as $h^k$, the hidden size $D$, head dimension as $d$, intermediate size $I$, and vocabulary size $V$.

$$F^{Att} = F^{qkv} + F^{qk} + F^{softmax} + F^{av} + F^{out}$$
$$F^{qkv} = 2 \times s \times D \times d \times h^q + 2 \times 2 \times s \times D \times d \times h^k$$
$$F^{qk} = 2 \times h^q \times s \times (s + s^{pst}) \times d$$
$$F^{softmax} = h^q \times (s + s^{pst}) \times (s + s^{pst})$$
$$F^{av} = 2 \times h^q \times s \times (s + s^{pst}) \times d$$
$$F^{out} = 2 \times s \times d \times h^q \times D \tag{12}$$

$$F^{Oth} = F^{up} + F^{gate} + F^{down} + F^{lm}$$
$$F^{up} = 2 \times s \times D \times 2 \times I$$
$$F^{gate} = s \times \times I$$
$$F^{down} = 2 \times s \times D \times I$$
$$F^{lm} = 2 \times s \times D \times V \tag{13}$$

## C ADDITIONAL EFFICIENCY ANALYSIS

We further evaluate the efficiency of Activation Beacon by decomposing the latency of pre-filling and decoding. Specifically, we set the context length to 32K and 128K and enforce the model to

[3]https://github.com/benbrandt/text-splitter

Table 6: Comparison of the training time and the peak GPU memory usage during training between Full-FT and Activation Beacon.

| Model | Method | DeepSpeed Stage | Training Time | GPU Memory |
|---|---|---|---|---|
| Qwen-2.5-7B | Full-FT | Zero-2 | 6.32 | 51.2 |
| | Ours | Zero-2 | 6.58 | 38.5 |
| Qwen-2.5-14B | Full-FT | Zero-3 (OOM w/ Zero-2) | 18.67 | 79.4 |
| | Ours | Zero-2 | 12.34 | 75.6 |

Table 7: Evaluation on RULER (Hsieh et al., 2024). Initially, Activation Beacon lags behind full-attention models on reasoning/aggregation tasks, yet the gap can be easily compensated by additional fine-tuning with synthetic 200 samples.

| Model | Method | NIAH AVG | VT | CWE | FWE | QA AVG |
|---|---|---|---|---|---|---|
| Qwen-2.5-7B | Full | 79.06 | 88.00 | 41.04 | 66.67 | 40.25 |
| | Full-FT | 80.13 | 71.95 | 32.28 | 64.76 | 52.38 |
| | Ours | 78.43 | 25.30 | 10.12 | 60.00 | 52.15 |
| | Ours + Synthetic FT | 80.91 | 85.30 | 59.30 | 72.18 | 51.27 |

generate precisely 128 tokens. We use a uniform x8 compression ratio. The peak GPU memory during the entire generation process is also reported.

According to Table 5, our method accelerates both pre-filling and decoding, and the acceleration extent amplifies as the context gets longer. Meanwhile, our method is better at speeding up decoding because it directly conditions on the beacon tokens' activations, which are 8x shorter than the raw activations used by the baseline. Moreover, Activation Beacon significantly reduces the peak GPU memory usage, enabling efficient processing of long context on customer GPUs.

Lastly, we also compare the training efficiency of Activation Beacon against the standard fine-tuning. The results are shown in Table 6. The experiment result indicates that Activation Beacon achieves comparable training speed as the Full-FT baseline while significantly reducing the memory cost.

## D  RULER EVALUATION

In addition to LongBench and NIAH, we evaluate the performance of Activation Beacon on RULER (Hsieh et al., 2024), a challenging long-context benchmark that consists of 13 synthetic tasks. Concretely, there are 6 tasks extending the regular NIAH to multi-key, multi-value, and multi-query variants; 1 task (Variable Tracking) aiming at examining the multi-hop reasoning capability by tracking variable assignments; 2 tasks (Common/Frequent Words Extraction) targeting on the aggregation capability by counting the word occurrences; and 2 tasks performing question answering while inserting noisy context to the ground-truth passages. Our evaluation is based on 128K context length and x4 compress ratio for Activation Beacon. The results are shown in Table 7.

It can be observed that our initial Activation Beacon model (denoted as "Ours") maintains a competitive performance on QA tasks, while lagging far behind the full-attention baseline on reasoning/aggregation tasks (VT and CWE). A similar drop can also be observed on Full-FT, too. One likely reason for this disadvantage is that our current fine-tuning recipe only uses one-hop QA data (as stated in Appendix A), which does not teach the model to perform complex reasoning or counting, resulting in inferior performance on these tasks. However, this problem should be mitigated by adjusting the composition of training data. We add 200 synthetic samples (100 for VT and 100 for CWE) with 20K maximum context length to the training data and fine-tune the model. The new model, denoted as "Ours+Synthetic FT", achieves substantial improvements on both VT and CWE while preserving its competitive performances on other tasks. This observation further validates the effectiveness of our Activation Beacon: it can quickly learn the desired compression capability given a limited set of training data.

Table 8: Evaluation of 7B and 14B models on LongBench (Bai et al., 2023). Activation Beacon always maintains a comparable performance to the expensive full-attention fine-tuned baseline.

| Model | Method | Length | Single-Doc | Multi-Doc | Summ. | Few-Shot | Code |
|-------|--------|--------|-----------|-----------|-------|----------|------|
| Qwen-2.5-7B | Full | 32K | 41.9 | 45.2 | 26.5 | 69.1 | 64.9 |
| | Full-FT | 32K | 42.7 | 46.1 | 26.7 | 67.6 | 66.3 |
| | Ours | 32K | 42.5 | 45.8 | 26.8 | 67.4 | 66.4 |
| Qwen-2.5-14B | Full | 32K | 42.5 | 52.9 | 25.1 | 71.7 | 66.7 |
| | Full-FT | 32K | 43.9 | 50.5 | 27.1 | 68.8 | 67.1 |
| | Ours | 32K | 43.4 | 49.9 | 27.1 | 68.5 | 67.4 |

# E    EXPERIMENTS ON LARGER MODELS

We apply Activation Beacon on Qwen-2.5-7B and Qwen-2.5-14B to inspect its performance on larger models. The LongBench evaluation results (with a uniform x4 compression ratio) are shown in Table 8. It can be observed that Activation Beacon retains its effectiveness, holding comparable performance to the full-attention fine-tuned baseline.

Additionally, we report the training & inference efficiency of 14B models in Table 5 and Table 6, respectively. Activation Beacon significantly reduces the memory costs during training, enabling efficient training of large models at high speed. Besides, it stably accelerates the inference process while being much more memory-efficient than the full-attention baseline.

