# OpenReview forum: "Long Context Compression with Activation Beacon"
_ICLR.cc/2025/Conference — ICLR 2025 Poster_

### Official Review · Reviewer_4UYt · 2024-10-17

**Soundness:** 3
**Presentation:** 3
**Contribution:** 3
**Rating:** 8
**Confidence:** 3

**Summary:**

The paper introduces a method called Activation Beacon for efficient long-context processing. The method adds learned "beacon" tokens at regular intervals in the input query. These tokens are expected to learn "summaries" of the text. At inference time, when processing long contexts, the beacon tokens are retained and the other context tokens are discarded. Thus, the beacon tokens essentially provide a summary of the context. The authors evaluate their method in comparison with a few other recent methods for efficient long context processing. Their method significantly improves results on LongBench and Multi-Needle-in-a-Haystack. The authors also provide ablations for various design choices.

**Strengths:**

- The paper focuses on an impactful area (long-context efficiency for LLMs).
- The paper provides a relatively simple idea that is well-explained. I view simplicity as a plus - if a simple idea can give strong accuracy improvements, it's far better than an unnecessarily complicated idea.
- The paper demonstrates strong results. Table 2 demonstrates strong accuracy at good latency on standard benchmarks for long context. Their method is competitive with full fine-tuning and better than baselines. Table 1 provides strong accuracy as well (though latency is missing).
- The figures do a good job of explaining what's going on. Figure 1 and Figure 2 give nice overviews of the method.
- The method is computationally efficient compared to fine-tuning. Their "pretraining" (starting from an already-pretrained model) only requires 1B tokens which is very few.
- The paper ablates design choices (Table 4).
- The paper is generally well written.

**Weaknesses:**

- In Table 1, it's not obvious whether the latencies are comparable. The compression ratio isn't mentioned.
- line 368: why do you use adaptive compression for llama-2 and uniform compression for qwen?

My main perceived weaknesses are regarding differences with previous works, and understanding why this method is performing so well:
- line 135: "ICAE and AutoCompressor... segment the long context into chunks and compress each chunk. However, both of them compress the context into soft tokens" <- how are these soft tokens different than beacon tokens? (similarly, on line 373-374, you mention soft tokens being a drawback)
- line 137: "Their compression workflow also lacks fine-grained handling of the chunked inputs, resulting in inferior compression quality" <- it seems like all they would need to do to allow "fine-grained handling of the chunked inputs" is just choose a smaller chunk size, so that the soft tokens appear more frequently. Is that right?
- - If this is true, it seems like your main contribution is the insight that soft tokens should be distributed evenly through the context. Would doing this massively improve the accuracy of ICAE and AutoCompressor? It seems like this is the main discovery, but I'm left wondering if I'm missing some more fundamental difference.

[Minor]:
line 47: "it it" -> "it"
line 53: "alternamtive" -> alternative
line 371: "highligh" -> "highlight"
line 483: "scope" -> score
Table 2: give units of "latency"

**Questions:**

My main question is in the "regarding differences with previous works" above. I want to understand if the results are improved mainly from decreasing chunk size, or if there's another difference between soft tokens and beacon tokens that explains the difference.

Also, what window size do you use? From Table 1, your model has a context length of 32k. I'm guessing you use this window size, but I don't see it explicitly stated, and line 184 suggests that 1024 would be a common window size, so I'm not sure. Since LongBench has only a few examples above 32k, I'm guessing the window logic isn't really used much (unlike for Needle In a Haystack)

---

> ### Author Response · Authors · 2024-11-24
> **Response Part I**
>
> Dear Reviewer,
>
> Thank you very much for your thorough review and constructive feedback! We greatly appreciate the opportunity to address your questions with the following response.
>
> > In Table 1, it's not obvious whether the latencies are comparable. The compression ratio isn't mentioned.
> > line 368: why do you use adaptive compression for llama-2 and uniform compression for qwen?
>
> The setting of compression ratio is stated in Table 1 of our paper (In L368-L370). Here, we make the following additional clarification.
> - The adaptive compression ratio performs the minimum degree of compression that can just fit each input context into the LLM's context window, e.g., the adaptive compression ratio is set to x4 given an 16K input and 4K context window.
> - **The adaptive compression ratio allows us to make a direct comparison with a LLM of short context window**, e.g., the Llama-2 baseline which is limited by a 4K context window (i.e., the ``Full'' method based on Llama-2 in Table 1).
> - Since Qwen-2 already has a 32K context window, which is long enough to cover any input data in LongBench, we no longer need to make adaptive compression. As a result, we simply set a uniform compression ratio (x4) for all compression methods.
> - The detailed analysis of latency is presented in Table 2, where all methods rely on the same x8 compression ratio.
>
> > line 135: "ICAE and AutoCompressor... segment the long context into chunks and compress each chunk. However, both of them compress the context into soft tokens" <- how are these soft tokens different than beacon tokens? (similarly, on line 373-374, you mention soft tokens being a drawback)
>
> The following differences are highlighted for the two types of tokens.
>
> - The ``soft tokens'' compress the context as their outputs from LLM, which constitutes $M$ embeddings (#soft_tokens: $M$). To optimize the compression effect, the compression module needs to adjust these $M$ embeddings.
> - In contrast, ``beacon tokens'' (#beacon_tokens: $M$) directly compress the layer-wise KV activations of LLM. Since the LLM is made up of multiple layers (#layers: $N$) and each layer contains multiple heads (#heads: $H$), the compression effect is optimized by adjusting these $M\times N\times H$ embeddings.
> - Therefore, our method enjoy a much larger degree of freedom, making it easier to optimize the compression effect.
> - Besides, the compressed KV activations from beacon tokens can be directly utilized by LLM, which makes it more efficient than soft tokens that require re-encoding before generation.

---

> ### Author Response · Authors · 2024-11-24
> **Response Part II**
>
> > line 137: "Their compression workflow also lacks fine-grained handling of the chunked inputs, resulting in inferior compression quality" <- it seems like all they would need to do to allow "fine-grained handling of the chunked inputs" is just choose a smaller chunk size, so that the soft tokens appear more frequently. Is that right?
> > If this is true, it seems like your main contribution is the insight that soft tokens should be distributed evenly through the context. Would doing this massively improve the accuracy of ICAE and AutoCompressor? It seems like this is the main discovery, but I'm left wondering if I'm missing some more fundamental difference.
>
> - The fine-grained compression is indeed important in our method. According to our ablation study in Table 4, the proposed placement of beacon tokens leads to a +15% improvement of performance. The same placement strategy can also be helpful to the baseline methods.
> - However, as we mentioned in the previous response, AutoCompressor and ICAE are fundamentally different from our method given that AutoCompressor & ICAE compress the context into their output embeddings from LLM, while our method directly compress the context into the LLM's KV activations. As explained in our previous response, the KV compression is easier to optimize and more efficient than the baseline compression methods.
>
> > Also, what window size do you use? From Table 1, your model has a context length of 32k. I'm guessing you use this window size, but I don't see it explicitly stated, and line 184 suggests that 1024 would be a common window size, so I'm not sure. Since LongBench has only a few examples above 32k, I'm guessing the window logic isn't really used much (unlike for Needle In a Haystack)
>
> The corresponding terminologies are clarified as follows. We'll explain them more rigorously in the revised pdf to avoid any ambiguity.
>
> - The **Length** column of Table 1 indicates the input length, i.e. the number of raw tokens in the input, which is denoted as $n$ in our manuscript. The **window size**, or **chunk size**, refers to the number of raw tokens gathered in each forward pass of compression (there is a schematic illustration in Figure 1). It is fixed to 1024 for Llama-2 and 2048 for Qwen-2, denoted as $w$ in our manuscript.
>
> - The **context window size** refers to the maximum number of tokens the backbone LLM can process. For example, it is 4K for Llama-2 and 32K for Qwen-2, which is denoted as $N$ in our manuscript. Because our method compresses the raw input into a more compact forms, the LLM is enabled to perceive a longer input beyond its original context window size.

---

> ### Comment · Reviewer_4UYt · 2024-12-01
> **Thank You For the Reply**
>
> The authors have provided the necessary clarifications. I maintain my score.

---

### Official Review · Reviewer_9Nqc · 2024-10-28

**Soundness:** 3
**Presentation:** 3
**Contribution:** 3
**Rating:** 8
**Confidence:** 3

**Summary:**

This paper compresses activations (keys and values) rather than using soft prompts, facilitating a progressive, fine-grained compression process. Specifically, it first partition input into small chunks, interleaving special beacon tokens that accumulate contextual activations.

**Strengths:**

- The paper presents an efficient method to compress long contexts, reducing memory usage by up to 8x and speeding up inference by 2x.
- Its progressive, fine-grained compression approach maintains high compression quality, allowing the model to handle longer inputs than its built-in context window.
-It supports flexible compression ratios, preserving model performance across various long-context tasks without degrading short-context capabilities.

**Weaknesses:**

- Lack of Comparison with KIVI: The paper does not provide a direct comparison with KIVI, a relevant compression method that could offer insights into the performance trade-offs.
- GPU Time Omission: The paper does not report GPU training or inference time, leaving uncertainty around the practical computational cost and efficiency of the proposed method.
- Scalability Concerns: The method requires 8 A800 GPUs to train a 7B parameter model, raising concerns about its scalability to larger models like 70B, where computational demands could become prohibitive.
- Limited Comparative Analysis: The paper would benefit from including more baseline methods, particularly compression-based approaches like KIVI, KV-layer shared compression methods such as CacheGen, and relative-position encoding strategies like LM-Infinite.
Additional References Needed: Incorporating comparisons with relevant works, such as LM-Infinite [1] for dynamic context management, CacheGen [2] for efficient context loading, and KIVI [3] for asymmetric quantization of KV caches, would strengthen the evaluation and highlight the advantages and limitations of the proposed approach.

[1] LM-Infinite: Simple On-the-Fly Length Generalization for Large Language Models
[2] CacheGen: Fast Context Loading for Language Model Applications
[3] KIVI: A Tuning-Free Asymmetric 2bit Quantization for KV Cache

**Questions:**

overall, this paper is novel and idea is well presented. please add more techniques for comparison so that users can choose different method.

---

> ### Author Response · Authors · 2024-11-24
> **Response Part I**
>
> Dear Reviewer,
>
> Thank you very much for your thorough review and constructive feedback! We greatly appreciate the opportunity to address your questions with the following response.
>
> > Lack of Comparison with KIVI: The paper does not provide a direct comparison with KIVI, a relevant compression method that could offer insights into the performance trade-offs.
>
> KIVI is an excellet method that is important in reducing the size of KV cache size. However, we regard KIVI as an orthogonal study to our work because it compresses the numerical values of KV cache, while our method compresses the sequence length of KV cache.
>
> Considering that the KV cache is a 4-dimensional tensor, the compression can be performed along all four dimensions. For example,
> - CLA[1] compresses along the layer dimension by sharing KV cache across layers;
> - GQA[2] compresses along the head dimension by sharing keys and values across attention heads;
> - MLA[3] compresses along the channel dimension by learning down projection and up projection matrices;
> - Our method compresses along the sequence dimension by compressing raw KV into beacon tokens' KV.
>
> Theoretically speaking, the compression along the above four dimensions, as well as the numerical compression made by KIVI, can be integrated together. We deem this as an interesting and promising direction for future research.
>
> In the following table, we compare KIVI with our work under the same KV compression ratio (x8). We also report the latency and peak GPU memory measured on NarrativeQA. We have the following observations from the experiment result:
> - *Our method and KIVI are equally effective for compression*, as they maintain comparable generation quality to the uncompressed baseline.
> - *Our method is more effective at reducing the generation latency and GPU memory usage*. This is because KIVI needs to perform de-quantization before self-attention.
>
> |Method|Compression Ratio|Single-Doc QA|Multi-Doc QA|Summarization|Few-Shot|Code|Latency (s)|Peak GPU Memory (G)|
> |:-:|:-:|:-:|:-:|:-:|:-:|:-:|:-:|:-:|
> |Llama-2-7B-FT|--|34.8|27.5|23.2|61.8|57.8|1.1915|34.6|
> |KIVI (2bit R32)|x8|34.5|27.1|22.8|61.5|57.6|0.6403|22.4|
> |Activation Beacon|x8|34.1|26.9|24.0|61.0|57.6|0.6019|20.7|
>
> > GPU Time Omission: The paper does not report GPU training or inference time, leaving uncertainty around the practical computational cost and efficiency of the proposed method.
> - The inference efficiency is discussed in Section 4.3 of the paper. Notably, Activation Beacon reduces the latency by 2 times when using a x8 compression ratio.
> - In addition, we compare the training throughput of Activation Beacon against the direct fine-tuning of full-attention baseline (Full-FT). It can be observed that the total training time of Activation Beacon is similar to the baseline, indicating that the extra training cost is very small.
> |Method|Overall Training Time (h)|Throughput (tokens/s)|
> |:-:|:-:|:-:|
> |Full-FT|6.32|43.9K|
> |Activation Beacon|6.58|42.2K|
>
> > Scalability Concerns: The method requires 8 A800 GPUs to train a 7B parameter model, raising concerns about its scalability to larger models like 70B, where computational demands could become prohibitive.
>
> - Although we use a 8xA800 GPU machine to accelerate the training process (a common configuration in related studies), the minimum training requirement is much smaller. In fact, the model can be trained with one single GPU of less than 40GB memory.
>
> - Additionally, our methods only consumes 2B training tokens. In contrast, other closely related baselines, like AutoCompressors, and CEPE, call for much more training tokens (e.g., 20B).
>
> Because of these features, our method preserves a small training cost, making it suitable for the application to larger models. To demonstrate this point, we compare the training time and GPU VRAM usage when training Qwen-2.5-7B and Qwen-2.5-14B. The experiment result indicates that Activation Beacon achieves comparable training speed as the Full-FT baseline while significantly reducing the memory cost.
>
> |Method|DeepSpeed Stage|Training Time (h)|Training GPU VRAM (G)|
> |:-:|:-:|:-:|:-:|
> |Full-FT (7B)|Zero-2|6.32|51.2|
> |Beacon (7B)|Zero-2|6.58|38.5|
> |Full-FT (14B)|Zero-3 (OOM w/ Zero-2)|18.67|79.4|
> |Beacon (14B)|Zero-2|12.34|75.6|
>
>
> [1] Cross-Layer Attention. https://arxiv.org/abs/2405.12981
>
> [2] Grouped Query Attention. https://arxiv.org/pdf/2305.13245
>
> [3] DeepSeek-V2. https://arxiv.org/abs/2405.04434

---

> ### Author Response · Authors · 2024-11-24
> **Response Part II**
>
> > Limited Comparative Analysis: The paper would benefit from including more baseline methods, particularly compression-based approaches like KIVI, KV-layer shared compression methods such as CacheGen, and relative-position encoding strategies like LM-Infinite. Additional References Needed: Incorporating comparisons with relevant works, such as LM-Infinite [1] for dynamic context management, CacheGen [2] for efficient context loading, and KIVI [3] for asymmetric quantization of KV caches, would strengthen the evaluation and highlight the advantages and limitations of the proposed approach.
>
> Thanks a lot for pointing out these interesting methods! We will update our discussions about related works accordingly.
>
> As mentioned in our previous response, KV compression is a very broad topic. Generally speaking, it can be performed from five dimensions. For example, CLA from the layer dimension, GQA from the head dimension, MLA from the channel dimension, KIVI from the numerical dimension, and our work from the sequence dimension. Besides, there are other alternative strategies, as presented by LM-Infinite and CacheGen.
>
> In this work, we've demonstrated our effectiveness in comparison with the existing sequence-level compression baselines. Our method also achieves equally competitive performance as KIVI under the same compression ratio while being more efficient. We believe compression strategies from different perspectives are complementary to each other.

---

> ### Comment · Reviewer_9Nqc · 2024-11-26
>
> Most of my concerns has been addressed properly. I would like to increase the score.

---

### Official Review · Reviewer_JN1J · 2024-11-04

**Soundness:** 3
**Presentation:** 2
**Contribution:** 3
**Rating:** 6
**Confidence:** 4

**Summary:**

The paper introduces "Activation Beacon", a compression method designed to enhance long-context processing efficiency in LLMs. The approach compresses the activations of keys and values in transformer layers, avoiding bottlenecks associated with traditional soft prompt methods. Additionally, a progressive compression workflow compresses each context unit in chunks, allowing the model to handle longer contexts than the original LLM's window. Experimental results show Activation Beacon achieves significant memory and computation savings, with minimal loss in performance.

**Strengths:**

1. Activation Beacon reduces inference time by 2x and KV cache memory costs by 8x compared to the uncompressed baseline.

2. The method supports adaptive compression ratios, allowing flexibility for different tasks and contexts.

3. The proposed model maintains short-context capabilities, preserving the performance of the original LLM.

**Weaknesses:**

1. The performance of this method may vary with model size. Current evaluations focus on medium-sized models, lacking validation on larger-scale models, leaving its effectiveness and applicability in very large models underexplored.

2. The added complexity of managing beacon tokens and compression ratios increases implementation overhead for end-users, particularly when adapting to different tasks. In addition to actual inference latency, specific memory usage data across implementations would help clarify practical resource requirements.

**Questions:**

See weaknesses

---

> ### Author Response · Authors · 2024-11-24
> **Response**
>
> Dear Reviewer,
>
> Thank you very much for your thorough review and constructive feedback! We greatly appreciate the opportunity to address your questions with the following response.
>
> > The performance of this method may vary with model size. Current evaluations focus on medium-sized models, lacking validation on larger-scale models, leaving its effectiveness and applicability in very large models underexplored.
>
> - In our paper, we focus on 7B models because it is the common setting for most baseline methods. This choice allows us to have a fair comparison with the baselines.
> - The proposed method is not limited to small models. To verify this point, we perform additional investigations with a larger LLM: Qwen-2.5-14B. The experiment result is shown in the following table.
> - According to our result, Activation Beacon retains its effectiveness when applied to the larger model, as it significantly outperforms the original LLM (Qwen-2.5-14B) and maintains a comparable performance as the expensive fine-tuned full-attention baseline (Qwen-2.5-14B-FT).
>
> |Method|Single-Doc QA|Multi-Doc QA|Summarization|Few-Shot|Code|
> |:-:|:-:|:-:|:-:|:-:|:-:|
> |Qwen-2.5-7B|41.9|45.2|26.5|69.1|64.9|
> |Qwen-2.5-7B-**FT**|42.7|46.1|26.7|67.6|66.3|
> |Qwen-2-7B-**Beacon**|42.5|45.8|26.8|67.4|66.4|
> |Qwen-2.5-14B|42.5|52.9|25.1|71.7|66.7|
> |Qwen-2.5-14B-**FT**|43.9|50.5|27.1|68.8|67.1|
> |Qwen-2.5-14B-**Beacon**|43.4|49.9|27.1|68.5|67.4|
>
> > The added complexity of managing beacon tokens and compression ratios increases implementation overhead for end-users, particularly when adapting to different tasks. In addition to actual inference latency, specific memory usage data across implementations would help clarify practical resource requirements.
>
> - To facilitate people's usage, we have encapsulated our method within the end-to-end generation function of huggingface `trasnformers`. It is very convenient to use, and users do not need to specify any extra parameters by themselves. Please check our [anonymous code](https://anonymous.4open.science/r/activation-beacon-anonymous-7875/README.md) for more details.
> - We study the efficiency of Activation Beacon with the following table. According to our result, Activation Beacon reduces the Peak GPU VRAM by 2.6 times, meanwhile accelerating the inference by 2 times.
>
> |Method|Context Length|Latency (s)|Peak GPU VRAM (G)|
> |:-:|:-:|:-:|:-:|
> |Qwen-2-7B|128K|4.399|64.3|
> |Qwen-2-7B-Beacon|128K|2.445|24.6|

---

> > ### Author Response · Authors · 2024-11-28
> >
> > Dear Reviewer JN1J,
> >
> > We have updated our submission with the inclusion of the following key results in our response.
> > - The investigation of Activation Beacon's effectiveness with larger models are presented in Table 8 (line 983).
> > - The analysis of peak memory requirement is reported in Table 5 (line 919).
> >
> > We've also provided anonymous source code to demonstrate the simplicity of using our method. Please feel free to let us know if there are any further questions about these issues.
> >
> > Thanks, \
> > The authors

---

> ### Author Response · Authors · 2024-11-30
> **Kind Request for Feedback**
>
> Dear ReviewerJN1J,
>
> We are deeply grateful for your valuable insights on our paper. With the discussion period closing in a few day, we would greatly appreciate your thoughts on our recent response. Your feedback is vital to ensure we have addressed your concerns adequately.
>
> Thank you for your time and consideration!
>
> Thanks,
> The authors

---

> ### Comment · Reviewer_JN1J · 2024-12-01
>
> Thanks to the author's reply. My main concerns are solved, so I raised the score.

---

### Official Review · Reviewer_ft7Q · 2024-11-05

**Soundness:** 2
**Presentation:** 2
**Contribution:** 3
**Rating:** 6
**Confidence:** 5

**Summary:**

The paper introduces “Activation Beacon,” a plug-in module to conduct long-context compression for LLMs. The proposed approach progressively compresses the activations at each layer and can be trained in the conventional auto-regressive way of language modeling. The authors demonstrate the benefits of this approach through evaluations on various long-context tasks for compression quality and inference efficiency.

**Strengths:**

- Compressing by chunks at each layer avoids the need for recomputation and addresses gradient back-propagation challenges present in some prior baselines that rely on recursive dependencies from final-layer outputs. This design enhances both training and inference efficiency.
- The chunking approach and the interleaved insertion of beacon tokens are straightforward and intuitive.
- Evaluations on various benchmarks indicate that the proposed approach generally outperforms the KV cache compression and “soft-prompt” compression baselines, achieving notable reductions in both inference time and memory usage.
- Training with randomly sampled compression ratios enables flexible compression ratios during testing.

**Weaknesses:**

- In addition to LongBench and NIAH, it is essential to evaluate the proposed approach on newer, more challenging benchmarks, such as RULER [1].
- Some recent context compression baselines, including CEPE [2] and LLoCO [3], are not discussed in the paper and should be included for a more comprehensive discussion or comparison.

[1] Hsieh et al. RULER: What's the Real Context Size of Your Long-Context Language Models? COLM 2024.
[2] Yen et al. Long-Context Language Modeling with Parallel Context Encoding. ACL 2024.
[3] Tan et al. LLoCO: Learning Long Contexts Offline. EMNLP 2024.

**Questions:**

- How are rotary embeddings managed for the beacon tokens? Although the LLM processes a fixed chunk at a time, the relative positions of the beacon tokens vary across chunks. How are positional embeddings applied in these cases?
- Additional parameters are added and fine-tuned for self-attention projections specific to the beacon tokens. What is the impact of these added parameters on VRAM usage and latency? If the cost is significant, could LoRA fine-tuning be effective for the proposed activation beacons approach?
- What portion of time is allocated to prefilling and decoding? While the proposed method reduces some recomputation, it may require customized attention masks or iterative context processing, which could lack efficient kernel implementation or result in extra kernel calls. Please provide a latency breakdown of prefilling and decoding for specific workloads (e.g., 32/128k context, 128 decoded tokens) and compare it with the flash attention full-context baseline.
- How does the proposed approach affect fine-tuning throughput? Please compare its performance with Full-FT.

I am open to adjusting my ratings if all concerns and questions are adequately addressed.

---

> ### Author Response · Authors · 2024-11-24
> **Response Part I**
>
> Dear Reviewer,
>
> Thank you very much for your thorough review and constructive feedback! We greatly appreciate the opportunity to address your questions with the following response.
>
> (The experiments on RULER are still ongoing, but progress has been relatively slow due to recent resource constraints. We will provide an update on the results as soon as the experiments are completed.)
>
> > Some recent context compression baselines, including CEPE and LLoCO, are not discussed in the paper and should be included for a more comprehensive discussion or comparison.
>
> Thank you for point out these recent methods! Both works offered important insights in context compression, therefore, we will include them in our revised manuscript. Here are brief highlights about their differences with our method.
>
> - **CEPE** introduces a *standalone encoder* to compress the context into token embeddings. The compression result are used by LLM through an additional *cross-attention module*. Therefore, CEPE introduces extra overhead during the cross-attention operation. CEPE also calls for a substantial training cost. According to the original paper, it takes 20B tokens for pre-training and 10B tokens for instruction tuning, whilst our method only consumes 1/10 of the training tokens.
>
> - **LLoCO** is a *retrieval-based framework* to tackle long-context problems, consisting of a retrieval system, a compressor (which is AutoCompressors by default) and a decoder. There are two major differences with our work: 1. LLoCO relies on a retrieval system, 2. LLoCO calls for in-domain fine-tuning and a retrieval system. As stated in its paper, developing more effective compressor (the focus of our work) is orthogonal to their research.
>
> We also compare Activation Beacon with CEPE on LongBench using its official checkpoint (LLoCO's model weights are still not public-available. Besides, it calls for retrieval and in-domain fine-tuning, which is not directly comparable with other methods).
>
> |Method|Single-Doc QA|Multi-Doc QA|Summarization|Few-Shot|Code|
> |:-:|:-:|:-:|:-:|:-:|:-:|
> |CEPE|24.2|23.2|21.2|60.5|46.5|
> |Llama-2-7B-Beacon|34.9|27.5|25.0|61.4|57.8|
>
> Both methods leverage Llama-2-Chat as their backbone LLMs. The results indicate the our method achieves a better performance than CEPE despite its simpler structure and lower training cost.
>
> > How are rotary embeddings managed for the beacon tokens? Although the LLM processes a fixed chunk at a time, the relative positions of the beacon tokens vary across chunks. How are positional embeddings applied in these cases?
>
> - The positional embedding is applied based on each token's relative position in each chunk and its preceding beacon tokens.
> - The relative position is calculated as the summation of the number of preceding raw tokens in the current chunk and the number of beacon tokens from preceding chunks.
> - Consider the following example. Given a  chunk size 2048 and compression ratio x8, there will be  $2048/8=256$ beacon tokens produced for each chunk. If we have three chunks, the relative positions for the 3rd chunk are assigned as follows:
>
> $$[\underset{\text{beacon tokens of 1st chunk}}{\underline{0,1,\dots,255}},\quad\underset{\text{beacon tokens of 2nd chunk}}{\underline{256,257,\dots,511}},\quad\underset{\text{raw and beacon tokens of 3rd chunk}}{\underline{512,513,\dots,2815}}].$$

---

> ### Author Response · Authors · 2024-11-24
> **Response Part II**
>
> > Additional parameters are added and fine-tuned for self-attention projections specific to the beacon tokens. What is the impact of these added parameters on VRAM usage and latency? If the cost is significant, could LoRA fine-tuning be effective for the proposed activation beacons approach?
>
> - Given the adoption of grouped query attention by the existing LLMs, the size of key/value projection matrices are significantly reduced. Therefore, *our method only introduces a small amount of additional parameters*. For example, there are only 462M new parameters for our implementation with Qwen-2-7B.
>
> - As mentioned, the size of new parameters is very small, and the new parameters are not involved in any heavy computation that expands memory usage. In fact, there is merely 1GB extra GPU memory by these new parameters during the entire computation flow.
>
> - The new QKV matrices substitute the original QKV matrices only when beacon tokens are encoded. The corresponding operations can be easily implemented by the `scatter` operation in O(1) complexity. Therefore, there is very little impact on latency.
>
> - We report the latency and GPU VRAM usage of Qwen-2-7B, Qwen-2-7B-Beacon, and Qwen-2-7B-Beacon w/o Additional Params in the following table (which uses the LLM's original parameters to encode raw tokens and beacon tokens). *The experiment results verify that additional parameters introduces very little latency and VRAM usage.*
> |Method|Context Length|Latency (s)|Peak GPU VRAM (G)|
> |:-:|:-:|:-:|:-:|
> |Qwen-2-7B|128K|4.399|64.3|
> |Qwen-2-7B-Beacon w/o Additional Params|128K|2.441|23.7|
> |Qwen-2-7B-Beacon|128K|2.445|24.6|
>
> In a conclusion, the current implementation has been efficient enough. While there will be even less parameters with LoRA, there won't be too much room for further reduction of latency and memory consumption.
>
> > What portion of time is allocated to prefilling and decoding? While the proposed method reduces some recomputation, it may require customized attention masks or iterative context processing, which could lack efficient kernel implementation or result in extra kernel calls. Please provide a latency breakdown of prefilling and decoding for specific workloads (e.g., 32/128k context, 128 decoded tokens) and compare it with the flash attention full-context baseline.
>
> - Our method performs standard causal attention (refer to L245), which doesn't rely on customized attention masks. The kernel optimization of FlashAttention is directly applied in our approach (refer to L316).
> - Our current implementation iteratively processes all chunks, however, no overhead is observed compared with parallel processing. This is in line with recent researches that adopt iterative processing to reduce peak GPU usage [1].
> - We provide a latency breakdown of prefilling and decoding in the following table. It can be observed that *our method accelerates both pre-filling and decoding*, and the acceleration extent amplifies as the context gets longer. Meanwhile, our method is better at speeding up decoding because it directly conditions on the beacon tokens' activations, which are 8x shorter than the raw activations used by the baseline.
>
> |Method|Input Length|Output Length|Prefilling Latency (s)|Decoding Latency (s)|Total Latency (s)|
> |:-:|:-:|:-:|:-:|:-:|:-:|
> |Qwen-2-7B|32K|128|0.522|0.709|1.231|
> |Qwen-2-7B-**Beacon**|32K|128|0.514|0.506|1.020|
> |Qwen-2-7B|128K|128|3.312|2.081|5.393|
> |Qwen-2-7B-**Beacon**|128K|128|2.038|0.550|2.588|
>
>
> [1] MINI-SEQUENCE TRANSFORMER https://arxiv.org/pdf/2407.15892
>
> > How does the proposed approach affect fine-tuning throughput? Please compare its performance with Full-FT.
>
> We compare the overall fine-tuning time (roughly 1B tokens with 20K maximum context length) as well as the throughput of Activation Beacon and Full-FT. The results are summarized in the following table. It can be observed that Activation Beacon achieves competitive training throughput against Full-FT. Note that Full-FT cannot generalize well to context much longer than its training length, while Activation Beacon can.
>
> |Method|Overall Training Time (h)|Throughput (tokens/s)|
> |:-:|:-:|:-:|
> |Full-FT|6.32|43.9K|
> |Activation Beacon|6.58|42.2K|

---

> ### Author Response · Authors · 2024-11-26
>
> Dear Reviewer ft7Q,
>
> We have completed the experiment on RULER, whose results are reported in the following table.
>
> - First, the original Qwen-2-7B-Beacon maintains a competitive performance on QA tasks. However, its performance on CWE and VT tasks are lagging behind.
> - One likely reason for this disadvantage is that Qwen-2-7B-Beacon is mainly fine-tuned by QA-style data, which results in the performance decay in other unrelated tasks. However, such a problem should be mitigated by adjusting the composition of training data.
> - To verify the above conjecture, we add merely synthetic 200 VT and CWE samples to the training data and fine-tune the model, denoted as ``Qwen-2-7B-Beacon+Synthetic FT''. According to the following result, the new model achieves substantial improvements on both VT and CWE while preserving its competitive performances on other tasks.
>
> Method | NIAH AVG| Variable Tracking | Common Words Extraction | Frequent Words Extraction | QA AVG
> ---------|---------|---------|---------|---------|---------
> Qwen-2-7B	|79.06	|88.00	|41.04	|66.67	|40.25
> Qwen-2-7B-FT	|80.13	|71.95	|32.28	|64.76	|52.38
> Qwen-2-7B-Beacon	|78.43	|25.30	|10.12	|60.00	|52.15
> Qwen-2-7B-Beacon+Synthetic FT	|80.91	|85.30	|59.30	|72.18	|51.27

---

> > ### Author Response · Authors · 2024-11-26
> >
> > Please feel free to let us know any questions about the additional results and their analysis. We are looking forward to engage with the reviewer for further discussion.

---

> ### Comment · Reviewer_ft7Q · 2024-11-26
> **Response to rebuttal**
>
> I would like to thank the authors for their thorough rebuttals. As paper revisions are permitted during the rebuttal period, I ask the authors to update the paper to include the related literature (CEPE, LLoCO) and incorporate critical new results (e.g., latency breakdown, RULER results, etc.) at this stage.

---

> > ### Author Response · Authors · 2024-11-28
> > **Submission is updated**
> >
> > Dear Reviewer ft7Q,
> >
> > We have updated our submitted paper with the inclusion of the following key results:
> > - The discussion of additional related literature is made in line line 145-161
> > - The latency breakdown is presented in Table 5 (line 919-931)
> > - The RULER results are presented in Table 7 (line 973-981)
> >
> > Please check our updated paper for more details. We are looking forward to address any further questions from the reviewer.
> >
> > Thanks, \
> > The authors

---

> > > ### Comment · Reviewer_ft7Q · 2024-11-28
> > > **Response**
> > >
> > > Thank you for your response and the updated draft. My concerns have been addressed and I raised my score.

---

### Meta-Review · Area_Chair_zQs7 · 2024-12-19

**Metareview:**

This paper proposes a context compression method for transformer-based LLMs. The method progressively compresses the key and value activations for all layers into beacon tokens. The paper evaluates its benefits for quality and efficiency in long-context tasks.

All reviewers agree that the paper has certain strengths, but they also raised a few questions and requests: (1) a discussion about the scalability, (2) a report on the inference speed, and (3) adding more benchmarks and related works. During the rebuttal, the authors properly answered those questions. Thus, the AC agrees that this paper is ready to be published at this conference.

**Additional Comments On Reviewer Discussion:**

The reviewers raised questions about the method's scalability and inference time analysis and requested additional benchmarks and discussion on related works. The authors addressed most of them by adding explanations and experimental results; consequently, some reviewers raised their scores.

---

### Decision · Program_Chairs · 2025-01-22

Accept (Poster)